# Assessing the Suitability of CHA_2_DS_2_-VASc for Predicting Adverse Limb Events and Cardiovascular Outcomes in Peripheral Artery Disease Patients with Percutaneous Transluminal Angioplasty: A Retrospective Cohort Study

**DOI:** 10.3390/biomedicines12061374

**Published:** 2024-06-20

**Authors:** Yu-Tsung Cheng, Fu-Lan Chang, Po-Hsien Li, Wen-Chien Lu, Chien-Shan Chiu

**Affiliations:** 1Cardiovascular Center, Taichung Veterans General Hospital, 1650 Section 4 Taiwan Boulevard, Xitun District, Taichung 40705, Taiwan; smallzebra99@gmail.com; 2Department of Nursing, Taichung Veterans General Hospital, 1650 Section 4 Taiwan Boulevard, Xitun District, Taichung 40705, Taiwan; fulan6568@gmail.com; 3Department of Food and Nutrition, Providence University, 200 Section 7, Taiwan Boulevard, Shalu District, Taichung City 43301, Taiwan; pohsien0105@gmail.com; 4Department of Food and Beverage Management, Chung-Jen Junior College of Nursing, Health Sciences and Management, 217, Hung-Mao-Pi, Chiayi City 60077, Taiwan; m104046@cjc.edu.tw; 5Department of Dermatology, Taichung Veterans General Hospital, 1650 Section 4 Taiwan Boulevard, Xitun District, Taichung 40705, Taiwan; 6Department of Post-Baccalaureate Medicine, College of Medicine, National Chung Hsing University, 145 Xingda Road, South District, Taichung 402, Taiwan; 7College of Biotechnology and Bioresources, Da-Yeh University, 168, University Road, Dacun, Changhua 51591, Taiwan

**Keywords:** peripheral artery disease (PAD), modified CHA_2_DS_2_-VASc risk (MCR) score, major adverse cardiovascular events (MACEs), major adverse limb events (MALEs), percutaneous transluminal angioplasty (PTA)

## Abstract

Patients with peripheral artery disease (PAD) are at high risk of major adverse limb events (MALEs) and major adverse cardiovascular events (MACEs). CHA_2_DS_2_-VASc is a prognostic score for atrial fibrillation stroke risk; however, no study has evaluated its predictive ability for MALEs and MACEs in PAD patients who underwent percutaneous transluminal angioplasty. We conducted a retrospective cohort study on patients from Taiwan with PAD. The patients were stratified into four risk groups based on their modified CHA_2_DS_2_-VASc score. Cox proportional hazard models, 10-fold cross-validation, and receiver operating characteristic (ROC) analyses were utilized to evaluate the predictive ability of CHA_2_DS_2_-VASc for MALEs, MACEs, and MALEs + MACEs. Kaplan–Meier analysis estimated the survival probability of the risk groups. CHA_2_DS_2_-VASc was found to be a significant predictor of MACEs (hazard ratio (HR) 3.52 (95% confidence interval (95% CI) 1.00–12.12; *p* = 0.048), HR 4.18 (95% CI 1.19–14.36; *p* = 0.023), and HR 5.08 (95% CI 1.49–17.36; *p* = 0.009), for moderate-, high-, and very high-risk groups, respectively), while for MALEs and MALEs + MACEs, significance was achieved only for the high-risk group using a univariate model. For the multivariate adjusted model, the score was found to be a significant predictor of MACEs for only the very high-risk group, with an HR of 4.67 (95% CI 1.03–21.09; *p* = 0.045). The score demonstrated an AUC > 0.8, good discrimination (c-index > 0.8), and good calibration for predicting MACEs. However, it failed to achieve good performance for predicting MALEs and MALEs + MACEs. Based on all of the findings, CHA_2_DS_2_-VASc could potentially serve as a risk stratification score for predicting MACEs in patients with PAD, but it failed to qualify as a good predictor for MALEs.

## 1. Introduction

Lower extremity peripheral artery disease (PAD) has a high incidence, affecting more than 230 million people globally [1,2]. It is a condition that is characterized by stenosis or occlusion of the arteries, thus reducing the flow of blood to the affected limb. People suffering from PAD are at a 10–15 times higher risk of major adverse cardiovascular events (MACEs) [3]. It is further associated with risk of major adverse limb events (MALEs) due to extensive atherosclerosis leading to tragic consequences, such as lower extremity amputation, acute limb ischemia (ALI), and death [4]. Adverse cardiovascular events include component heart failure, non-fatal reinfarction, hospitalization due to cardiovascular conditions, repeat percutaneous coronary intervention (PCI), coronary artery bypass grafting, unscheduled coronary revascularization, and all-cause mortality [5]. MALEs comprise major amputations and peripheral revascularization with eventual morbidity [6,7].

CHA_2_DS_2_-VASc is a cumulative score that is based on predefined criteria, where “C” stands for congestive heart failure (CHF); “H” stands for hypertension (HTN); “A_2_” stands for age doubled > 75; “D” stands for diabetes mellitus (DM); “S_2_” stands for stroke (doubled), transient ischemic attack, or thromboembolism; “V” stands for vascular disease; “A” stands for an age range of 65–74 years old; and “Sc” stands for sex category (female). This score is commonly used for the risk stratification of strokes in patients with atrial fibrillation (AF) and is used to assist in decision making regarding anticoagulation therapy for stroke prophylaxis [8,9]. In recent studies, the CHA_2_DS_2_-VASc has been demonstrated to be a predictor of adverse clinical outcomes associated with coronary artery disease, stroke, and other cardiovascular conditions, regardless of AF [10,11]. A study reported that the CHA_2_DS_2_-VASc score had a high correlation with mortality in PAD patients and may therefore be useful as a predictor for the identification of high-risk PAD patients [11,12]. However, there are no studies that have evaluated or validated the predictive ability of CHA_2_DS_2_-VASc for MALEs and MACEs in PAD patients. This study utilized a modified CHA_2_DS_2_-VASc score for predicting the risk of MACEs, MALEs, and MALEs + MACEs in patients with PAD from Taiwan. Comprehensive analyses were conducted to evaluate the feasibility and efficacy of CHA_2_DS_2_-VASc as a predictive score for MALE and MACE outcomes in patients with PAD. CHA_2_DS_2_-VASc has been proven to have clinical applicability for stroke risk stratification in patients with AF. Determining whether it can be used for MALE and MACE risk stratification in patients with PAD who have undergone percutaneous transluminal angioplasty (PTA) was the aim of this study. 

## 2. Methods

### 2.1. Data Description

Demographic data along with the medical history of 601 Taiwanese subjects with and without PAD were obtained from Taichung Veterans General Hospital. Participants were enrolled from 2015 to 2020, and were followed up with until April 2022, with a median follow-up period of 662 days. Patients fulfilling the following criteria were excluded from the study: (i) patients aged <18 and >85 years old; (ii) pregnant women; (iii) patients with any cancer; (iv) patients with any infection (except for local wound infection in the lower limbs) at the time of recruitment; (v) patients with any extremity artery disease (except for lower extremity artery disease); and (vi) patients with missing information on PAD (Figure 1). A total of 503 patients with PAD were retained for analyses. Patients were considered to have PAD if they exhibited clinical presentation, such as an ankle brachial index (ABI) < 0.9, critical limb ischemia, intermittent claudication, resting pain, trophic changes, previous PTA, or Rutherford classification 1–6 (a classification system for PAD patients) (Table 1 and Appendix A). Indications for PTA treatment included the revascularization of multi-region vessels.

Clinical information and environmental exposures were recorded, including smoking history (ever- and never-smokers), hyperlipidemia (HPL), low-density lipoprotein (LDL) cholesterol, glycated hemoglobin (HbA1C), cholesterol, LDL, high-density lipoprotein (HDL) cholesterol, triglyceride (TG), glucose, CAD, a coronary artery bypass graft (CABG), percutaneous cardiac intervention (PCI), old myocardial infarction (MI), chronic obstructive pulmonary disease (COPD), chronic kidney disease (CKD) hemodialysis (HD) or peritoneal dialysis (PD), AF, autoimmune diseases, medications, and hospitalization records (details provided in the Appendix A). 

This retrospective study was approved by the Institutional Review Board of Taichung Veterans General Hospital (TCVGH-IRB#: CE21519A). All research was performed in accordance with the relevant guidelines/regulations of the ethical committee. The IRB committees I and II of Taichung Veterans General Hospital waived the need for informed consent. The patient information was de-identified, and the authors had no access to information that could identify individual participants during or after data collection. The data were accessed on 1 July 2022 for this research.

### 2.2. Endpoints and Events

The baseline time point was set at the primary intervention for all patients with PAD. The primary intervention consisted of PTA, which involved vascularization to unclog the artery, thus allowing blood to flow to the lower limbs of the patients suffering from PAD. Patients were followed up to three endpoints: (1) MALEs, (2) MACEs, and (3) MALEs and MACEs. MALEs were defined as repeat vascularization or amputation. MACEs were defined as non-fatal stroke, non-fatal myocardial infarction, or cardiovascular death. All patients’ information was followed up by telephone, medical chart, and clinical visit regarding non-fatal myocardial infarction. 

### 2.3. Statistical Analysis

#### 2.3.1. Modified CHA_2_DS_2_-VASc Risk (MCR) Score

First, the traditional CHA_2_DS_2_-VASc score of all patients was calculated by summing up the abnormalities defined based on whether the patients fulfilled one or more of the following criteria: CHF defined via chart review; ICD-10 code diagnosis; echocardiographic ejection fraction; HTN with a systolic blood pressure ≧ 140 mmHg; diastolic blood pressure ≧ 90 mmHg or on HTN medication; age ≥ 75 years; DM with either a prior diagnosis or recorded fasting blood sugar level ≧ 126 mg/dL; on anti-DM medication or newly diagnosed at the time of hospitalization; previous stroke; vascular disease defined as having PTA; age 65–74 years; and female gender. The CHA_2_DS_2_-VASc ranged from 0 to 9 for the study cohort, with very few patients demonstrating scores of 0–3 and 7–9. Therefore, to avoid bias and obtain a better distribution of risk scores, we used a modified CHA_2_DS_2_-VASc score (which for convenience purposes will be referred to as MCR from here on) that ranged between 3 and 6 by redefining the patients with 3 or fewer abnormalities as low-risk patients (MCR = 3), and those with 6 or more abnormalities as very high-risk patients (MCR = 6). The two intermediate groups were defined as moderate-risk (MCR = 4) and high-risk patients (MCR = 5), respectively (Appendix A). 

#### 2.3.2. Statistical Analysis Using CHA_2_DS_2_-VASc Risk Score as Predictor

Univariate and multivariate adjusted Cox proportional hazard regression models were fitted with MCR as the predictor for the following outcomes: (i) MALEs, (ii) MACEs, and (iii) MALEs + MACEs. Baseline variables such as hyperlipidemia, chronic obstructive pulmonary disease (COPD), chronic kidney disease, cardiac rehabilitation, and AF were selected as covariates for the adjusted multivariate models. Cox proportional hazard models [13] were employed using the “Survival” package in R to estimate the hazard ratio and 95% confidence intervals (CI) [14]. Kaplan–Meier analyses were conducted for all events to estimate the effect of the MCR score on the survival probability using the “Survival” package in R [14,15].

#### 2.3.3. Model Evaluation 

A 10-fold internal cross-validation was performed to conduct model discrimination and model calibration analyses [16,17,18]. Harrell’s c-index for discrimination analysis [17], which evaluates the concordance of predicted and observed survival and calibration analyses, was applied to compare the observed and predicted events among all study subjects by a given follow-up time. A random split of all study subjects using a 9:1 ratio for the training and testing sets, respectively, was performed for 10 repetitions. We further compared our MCR score-based models with traditional baseline models, but instead of using the MCR score, we utilized the traditional variables, such as CHF, HTN, DM, stroke, vascular disease, age, and sex, to predict the outcomes. The average c-indices across all 10 cross-validations were utilized to evaluate the proposed models’ performances for all three outcomes: (i) MALEs, (ii) MACEs, and (iii) MALEs + MACEs. The “Survival” package in R was used to conduct all analyses [14]. Furthermore, to evaluate the predictive ability of the CHA_2_DS_2_-VASc risk score, the R package “timeROC” was used to conduct receiver operating characteristic (ROC) analysis for MALEs, MACEs, and MALEs + MACEs at the time points of 12 months, 24 months, 36 months, and 48 months, respectively, and areas under the curves (AUCs) were obtained. 

## 3. Results

### 3.1. Baseline Demographic and Clinico-Pathological Variable Analysis

A total of 503 patients from a Taiwanese hospital cohort with PAD were included in this study. Patients with missing information (three patients) and those without PAD were excluded from this study (Figure 1). Table 1 provides a detailed account of the demographic and clinical characteristics of all patients included in the study analysis. Additionally, a detailed analysis was performed using all baseline demographic and clinical variables for PAD patients across the three study endpoints to see whether the association of the MCR with the incidence of the outcomes was affected by any confounders (Appendix A). The mean age of patients with MACEs was higher than that of patients without MACEs, while the mean age of MALE patients was slightly lower than that of those without MALEs. Furthermore, patients demonstrating MACEs had a significantly higher proportion of CHF, HTN, and DM when compared to patients without MACEs, but the trend was reversed in cases with a MALE. Hyperlipidemia was observed in a significantly higher number of patients with MACE (*p* = 0.05) outcomes when compared with those with no events. A complete description of the outcomes and the univariate analyses can be found in the Appendix A.

### 3.2. Modified CHA_2_DS_2_-VASc Risk (MCR) Score and Its Association with the Three Endpoints: MALEs, MACEs, and MALEs + MACEs

Table 2 summarizes the baseline characteristics of patients classified according to each of the MCR scores 3, 4, 5, and 6. The mean age of PAD patients increased with higher risk groups (59.14, 67.76, 72.2, and 79.6 years for MCR = 3, 4, 5, and 6, respectively). Increasing numbers of men were observed in groups with higher risks. CHF, HTN, DM, stroke, and vascular diseases all showed a rising trend with a higher MCR score. The prevalence of hyperlipidemia and ever-smokers was higher among low-risk groups in comparison to high-risk groups. 

Furthermore, for the MCR scores, the proportions of different events were explored to understand their associations with MALEs, MACEs, and MALEs + MACEs, as displayed in Figure 2. For MACE patients, a higher number of abnormalities was correlated with a slightly higher proportion of events (3%, 11.3%, 11.43%, and 11.49% of events are correlated with ≤3, 4, 5 and ≥6 abnormalities respectively); however, this correlation was slightly different for the other two events of MALEs (31%, 42.6%, 45%, 33.8%) and MALEs + MACEs (34%, 49.6%, 50%, 42.6%). The corresponding hazard ratio (HR) and 95% confidence interval (95% CI) are demonstrated via the forest plot in Figure 2.

Kaplan–Meier analysis using MCR for a maximum follow-up period of 56.5 months demonstrated that, for MACEs, the low-risk group (MCR = 3) had the longest time-to-event, followed by the moderate-risk, high-risk, and very high-risk groups (Figure 3b). The log-rank test displayed significance with a *p* value of 0.043. For MALEs (Figure 3a) and MALEs + MACEs (Figure 3c), MCR was not found to attain significance (*p* = 0.22 and 0.18, respectively) for predicting the time-to-event. 

The above findings indicate that the MCR could potentially act as a stratification score for identifying PAD patients that could be at a risk of MACEs, but the results for MALEs and MALEs + MACEs were not significant. 

### 3.3. Survival Analysis with the Modified CHA_2_DS_2_-VASc Risk (MCR) Score as a Predictor

Univariate and multivariate adjusted Cox proportional hazard regression was performed using the MCR score as the predictor for the endpoints of MALEs, MACEs, and MALEs + MACEs. The results are summarized in Table 3. The MCR score was found to be a significant predictor of MACEs using the univariate regression analysis with HR 3.52 (95% CI 1.00–12.12; *p* = 0.048), HR 4.18 (95% CI 1.19–14.36; *p* = 0.023), and HR 5.08 (95% CI 1.49–17.36; *p* = 0.009) for the moderate-risk, high-risk, and very high-risk groups, respectively (Table 3). For the multivariate adjusted regression analysis, significance was only observed at HR 4.67 (95% CI 1.03–21.09) *p* = 0.045* for the very high-risk group. The c-index values for predicting MACEs were 0.61 and 0.825, suggesting a good fit and a good discrimination ability for predicting MACEs. For the MALE outcome, MCR was a significant predictor only for the high-risk group using both univariate and multivariate regression analysis; univariate: HR 1.55 (95% CI 1.01–2.38; *p* = 0.046); and multivariate adjusted: HR 1.66 (95% 1.04–3.2; *p* = 0.039) (Table 3). The c-indices were also 0.54 and 0.56, respectively, demonstrating poor discrimination ability. Similarly, for the MALE + MACE outcome, both the univariate and multivariate adjusted regression demonstrated that MCR could significantly predict only the high-risk group (Table 3). Moreover, the c-indices were 0.54 and 0.58, respectively, showing poor discrimination ability. 

### 3.4. Evaluation of MCR as a Predictor for MALEs, MACEs, and MALEs + MACEs

A 10-fold cross-validation process was implemented to confirm the predictive performance of the MCR score and to confirm its reproducibility [16]. Table 4 lists the average and the standard deviations of the c-indices, which provide evidence that, for a given patient pair, MCR can effectively discriminate the occurrence of MACEs, which indicates that it is a good predictor. However, the discrimination power of MCR for MALEs and MALEs + MACEs was not good enough. Calibration analyses demonstrated that the differences in the proportions between the observed and predicted MACEs were ≤5% for the first 2 years, and for years 3–5, they were restricted to <10%, while for both MALEs and MALEs + MACEs, the average difference was quite high (33–49%), except for year 1 (<5%). Figure 4 and Appendix A showcase the detailed calibration results. Comparison with a traditional model (that used clinical parameters directly instead of the cumulative score) showed comparable results (Appendix A). 

ROC analyses at the time points of 12 months, 24 months, 36 months, and 48 months, respectively, were conducted (Figure 5). For MACEs, the univariate model and the multivariate adjusted model demonstrated a maximum AUC of ~0.66 and ~0.85 for the time points of 48 months and 12 months, respectively (Figure 5B,E), while for both the MALEs and MALEs + MACEs, the univariate model demonstrated AUCs of <0.6 (Figure 5A,C) for all time points, whereas the maximum AUC for the multivariate models was ~0.65 for time points > 36 months (Figure 5D,F). Based on all of our findings, we can reasonably say that the modified CHA_2_DS_2_-VASc is a good predictor of MACEs, but it does not qualify as a good predictor of MALEs and MALEs + MACEs for patients with PAD. Hence, the MCR can be used to conduct early risk stratification for MACEs for PAD patients with the goal of implementing treatments for secondary cardiovascular disease prevention.

## 4. Discussion

Peripheral artery disease is a narrowing of the peripheral arteries that carry blood away from the heart to other parts of the body and is associated with high rates of cardiovascular conditions and high rates of mortality [19]. The most common type of PAD is lower extremity PAD, in which blood flow is reduced to the legs and feet. Other forms of PAD, such as carotid artery stenosis, mesenteric artery stenosis, and upper-extremity PAD, are less common and may require different therapeutic strategies. This study focused on lower extremity PAD. 

A higher risk of cardiovascular events (MACEs) and limb events (MALEs) exist in patients with PAD [4]. Classification systems are important for selecting medical, surgical, and percutaneous treatment preferences. Various classification systems utilizing anatomical, clinical, and image data have been used for PAD in previous studies [19,20]. Recently, a score-based system, CHA_2_DS_2_-VASc, has become widely used for the effective grading of patients, providing physicians with objective criteria for patient assessment, treatment, and clinical follow-up of PAD and cardiovascular conditions [21]. 

This study utilized a modified version of the commonly used CHA_2_DS_2_-VASc score, the MCR score, and conducted comprehensive analyses to test and confirm its ability to predict the risk of incidence of MACEs and MALEs in patients suffering from PAD. Two regression models, univariate and multivariate adjusted, with MCR as the predictor, were fitted for three outcomes: MALEs, MACEs, and MALEs + MACEs. MCR was demonstrated to be a significant predictor of MACEs for moderate-risk, high-risk, and very high-risk patients, respectively, compared with the low-risk reference group patients (Table 3). However, the MCR was not very good at predicting MALEs (significant only for the high-risk group; Table 3) or MALEs + MACEs (significant only for high-risk group for the multivariate adjusted model; Table 3). Discriminant analysis and calibration analysis were conducted using 10-fold cross-validation, and AUCs were calculated for predicting the events for different time points. All results indicated CHA_2_DS_2_-VASc to be a suitable predictor of MACEs but not MALEs in patients with PAD. 

The CHADS_2_ score has been widely employed since 2001 for predicting the risk of stroke in patients with atrial fibrillation (AF) and demonstrates worse stratification performance compared with CHA_2_DS_2_-VASc [22]. This could be justified by the high prevalence rates of peripheral vascular diseases (PVDs) among AF patients, which were found to be associated with increased rates of mortality. Therefore, integrating the PVD incidence within the risk score improved its risk stratification. Hence, CHA_2_DS_2_-VASc was a better score for conducing risk stratification for patients with AF. 

Although the initial purpose of the scoring system was to predict the risk of stroke and MACEs in AF patients [23,24], over time, it has increasingly been used for various other stratification purposes [10]. For instance, different studies have used this risk score for different cardiovascular conditions, such as sick sinus syndrome, thromboembolism, and stroke [21,25,26,27]. Other recent studies have demonstrated an association between CHA_2_DS_2_-VASc and the risk of critical limb ischemia in peripheral arterial occlusive disease patients [10,28]. Another recent study utilized the CHA_2_DS_2_-VASc score to predict the risk of mortality in PAD patients with peripheral arteriography [12]. Due to the skewed distribution of the CHA_2_DS_2_-VASc score in our study cohort, a modified CHA_2_DS_2_-VASc risk score, MCR, was utilized in this study, which also demonstrated a high predictive ability for MACEs in all patients with PAD who underwent PTA. However, we failed to establish it as a good predictor for MALEs. 

PTA is a treatment strategy rendered to patients with lower extremity PAD in order to improve their lifestyle-hindering symptoms [26]. All of the patients included in this study demonstrated critical limb ischemia (CLI), and most of them had a long diffuse critical lesion in a leg vessel. The mainstay of treatment for CLI is to re-establish antegrade downstream flow in the leg. Therefore, most of the patients included in this study received revascularization of multi-region vessels. Usually, revascularization of an iliac lesion (common iliac artery + external iliac artery), femoropopliteal lesion (common femoral artery + superficial femoral artery + popliteal), or below-the-knee (BTK) lesion (peroneal + anterior tibial artery + tibioperoneal + posterior tibial artery + dorsalis pedis artery) is conducted concurrently, as it allows for longer survival with an increased quality of life compared to patients undergoing primary amputation [29]. Dual antiplatelet therapy (DAPT) was implemented 1–3 months after PTA. The major reason was to cover the period of stent re-endothelialization. All patients with atrial fibrillation were on warfarin/direct oral anticoagulant (DOAC), and >10% of patients were provided warfarin/DOAC treatment in both the MALE and non-MALE groups, according to the physician’s judgment, which was informed by reference data. This treatment was applied to balance the risk of ischemia and bleeding clinically. For identical reasons, anticoagulant management was also implemented in both the MACE group and the non-MACE group. 

This study provides valuable insights into the applicability of the MCR for the prediction of risk of limb events (MALEs) and cardiovascular events (MACEs) for PAD patients who underwent PTA, information on which was missing in the literature until now. The results for MALEs were not significant, and the performance of MCR as a predictor of MALEs was not encouraging (low discrimination ability, poor calibration, and low AUC). Possible reasons for this could be that MALEs occurred in roughly 35% of patients within the first year and were thereafter stable due to wound healing involving multiple factors, such as nutrition status, infection control, and wound debridement. This is in alignment with the COMPASS study, where the incidence of MALEs was 2.2% compared to the 6.9% incidence of MACEs. Even though MALEs are the outcome most feared by PAD patients, in reality, the incidence of MALEs is much lower than that of MACEs. 

Moreover, in a prior published review, the authors demonstrated the manner in which critical vascular events affect the lower limbs in subjects with PAD and stenosis greater than 70%, largely attributable not to the progression of the stenosis but to thrombotic phenomena [30]. Another similar study reviewed the role of coagulation in patients with PAD, thereby highlighting the importance of thrombotic phenomena in patients with arterial disease of the lower limbs [31]. 

We showed through this study that the MCR could successfully predict MACEs with a high discrimination ability and a high AUC. MACE outcomes in patients included non-fatal stroke, non-fatal myocardial infarction, and cardiovascular death and did not include procedure-related restenosis. Figure 2 shows that that the proportion of patients with MACEs rose steadily for higher-risk patients (i.e., patients with a higher number of abnormality parameters), while for MALEs, there was a drop in the proportion of events for the patients with the highest risk. Clinically, a higher score implies a higher risk of thromboembolism, which would prompt the treating physician to be more aggressive with anticoagulant or antiplatelet drugs [32]. This may explain why the high-risk patients with abnormal parameters demonstrated fewer MALEs, as the use of anticoagulants can significantly reduce acute limb ischemia in PAD patients after revascularization (Voyager study) [33]. Another possible reason could be the higher mortality rate in this group of patients, eventually resulting in relatively fewer MALEs.

On comparing the traditional model with the MCR-based regression model, MCR performed similarly or slightly better. The traditional model consists of multiple risk metrics that are associated with known caveats such as low statistical power, extreme higher-order interaction terms, low robustness, and collinearity among risk factors [34]. As cumulative scores such as MCR are summed across a number of variables, they possess the advantage of being a more stable measure and are more suitable for detecting effects as measurement errors are diminished when scores are summed [35]. This is why MCR is believed to be a more robust alternative to traditional models that could be used for risk stratification for MACEs of Taiwanese PAD patients who underwent PTA, thereby allowing for shared decision making. It is to be noted that CAD is an important comorbidity in PAD patients. A prior study on the REACH dataset demonstrated that one-third of the patients with CAD also had PAD, while almost two-thirds of PAD patients had a coexisting CAD or cerebrovascular disease, and the percentage of CAD in PAD patients was logically proportional to the ischemic risk [36]. Therefore, it is understandable that the MACE group had a higher burden of CAD than the non-MACE group. In addition to medicines, the C part (comorbidities and cardiovascular risk factor management including change in lifestyle) of the ABC pathway strategy, which is commonly adopted for AF patients, can also be selected for treating PAD patients with higher ischemic risk, based on risk stratification by MCR. 

The study population comprised patients with severe PAD (stenosis > 70%; Appendix A), demonstrating CLI with Rutherford classifications of 4–6. Based on the findings, CHA_2_DS_2-_VASc classification can be used to screen PAD patients, as early as possible, who are potentially at a moderate to very high risk of developing MACEs. Early diagnosis of PAD may allow physicians to prescribe dual pathway inhibition (DPI) treatment, as demonstrated through the COMPASS trial and Voyager trial [7,37]. DPI with rivaroxaban (novel oral anticoagulant (NOAC) drug) plus acetylsalicylic acid (ASA), commonly known as aspirin, can be used for secondary cardiovascular disease prevention, which would be beneficial for patients with PAD, who have been known to be at extremely high risk of developing MACEs, in comparison to the risk of MALEs [38]. Therefore, MCR can potentially be used to stratify PAD patients in a clinical setting so that preventive care and treatments could be adopted at the early stages of PAD for secondary cardiovascular outcome prevention.

One of the limitations of this study was the lack of a prospective external cohort to conduct validation of the performance of the MCR for predicting MACEs; however, a thorough internal validation was conducted instead. Moreover, we ensured that the analysis of the study cohort took into account the various clinical factors and the complexities associated with all events for PAD patients who underwent PTA. Hence, we believe that the findings could be generalized to specific subgroups of PAD patients. Nevertheless, future studies should be conducted to validate the findings using cohorts of independent external patients. The prognosis for PAD patients could vary based on the distribution of PAD lesions [39]. This study mainly focused on lower extremity PAD. Hence, future studies are needed to clarify the above. There were some other limitations in the data that were analyzed. No information on the number of stents and drug-eluting balloons (DEBs) was available. In addition, it was difficult to define the “target region” in CLI patients, as concurrent implementation of the revascularization of the iliac lesion, femoropopliteal lesion, or BTK lesion was conducted. Hence, whether the target region had any effect on the outcome could not be determined.

## 5. Conclusions

PAD confers an overall higher mortality risk as well as greater risks of coronary and cerebral ischemic events, and thus it is a condition with a high CVD risk. This necessitates strict preventive strategies. The findings of this study indicated that CHA_2_DS_2_-VASc could potentially serve as a risk stratification score for MACEs for a specific group of patients with lower extremity PAD who underwent PTA, thereby allowing medical practitioners to implement the appropriate therapeutic measures in a timely fashion to prevent worse MACE outcomes. The findings from this study also clearly indicate that CHA_2_DS_2_-VASc does not qualify as a good predictor of MALEs and therefore cannot be used for the risk stratification of patients with PAD for MALEs. The negation of the hypothesis for MALEs was obtained through extensive evaluation; therefore, we believe that this is valuable scientific information that was absent from the literature related to CHA_2_DS_2_-VASc up until now.

## Figures and Tables

**Figure 1 biomedicines-12-01374-f001:**
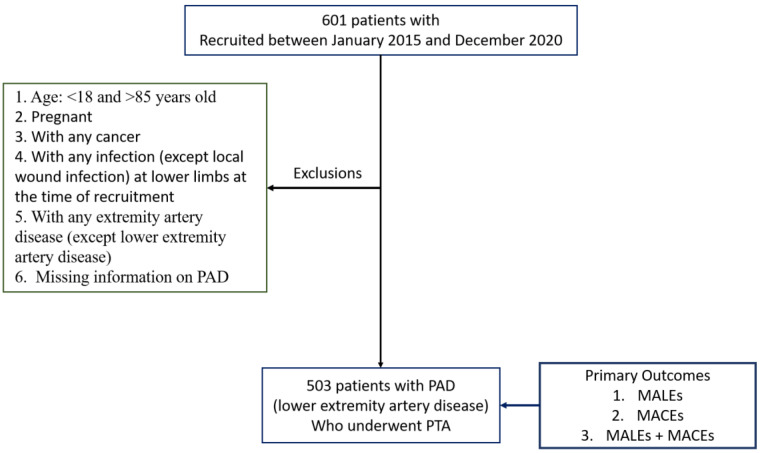
Exclusion criteria and inclusion of subjects for analyses.

**Figure 2 biomedicines-12-01374-f002:**
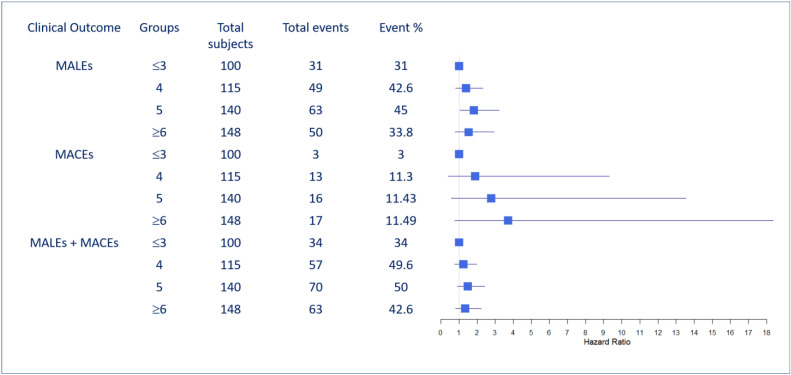
Event rates for each of the three outcomes, MALEs, MACEs, and MALEs + MACEs, for study subjects classified into four risk groups based on MCR risk parameters (low-risk, moderate-risk, high-risk, and very high-risk) (N = 503). MALEs: major adverse limb events; MACEs: major adverse cardiovascular events. Blue squares indicate hazard ratios and the blue line indicates the confidence intervals.

**Figure 3 biomedicines-12-01374-f003:**
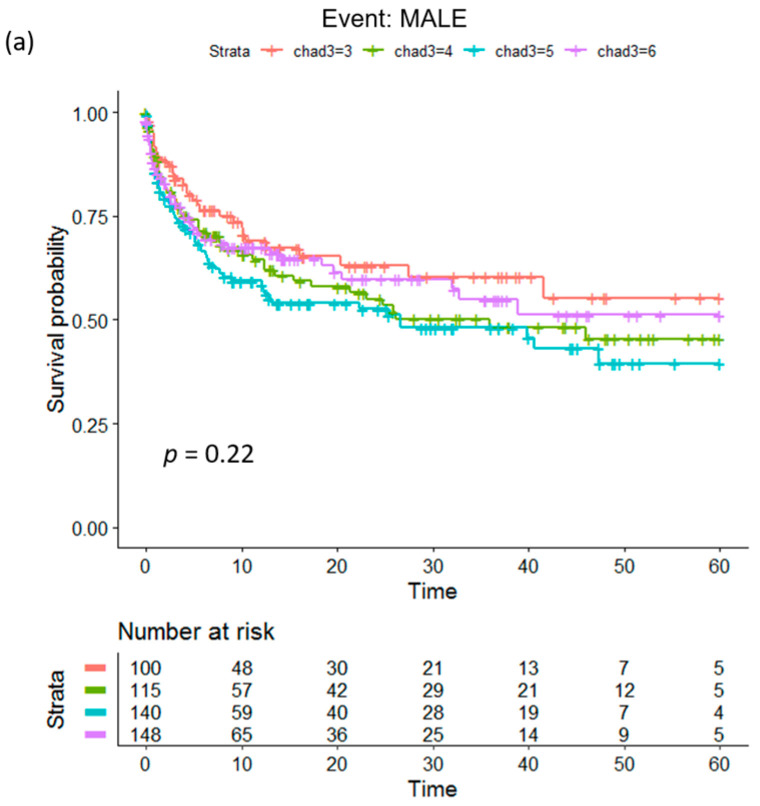
Kaplan–Meier plots to compare the time-to-event probability of subjects (N = 503) with different MCR scores (low-risk, moderate-risk, high-risk, very high-risk). The *p* values indicate whether significant differences exist among the different groups: (**a**) MALEs, (**b**) MACEs, and (**c**) MALEs + MACEs. MALEs: major adverse limb events; MACEs: major adverse cardiovascular events.

**Figure 4 biomedicines-12-01374-f004:**
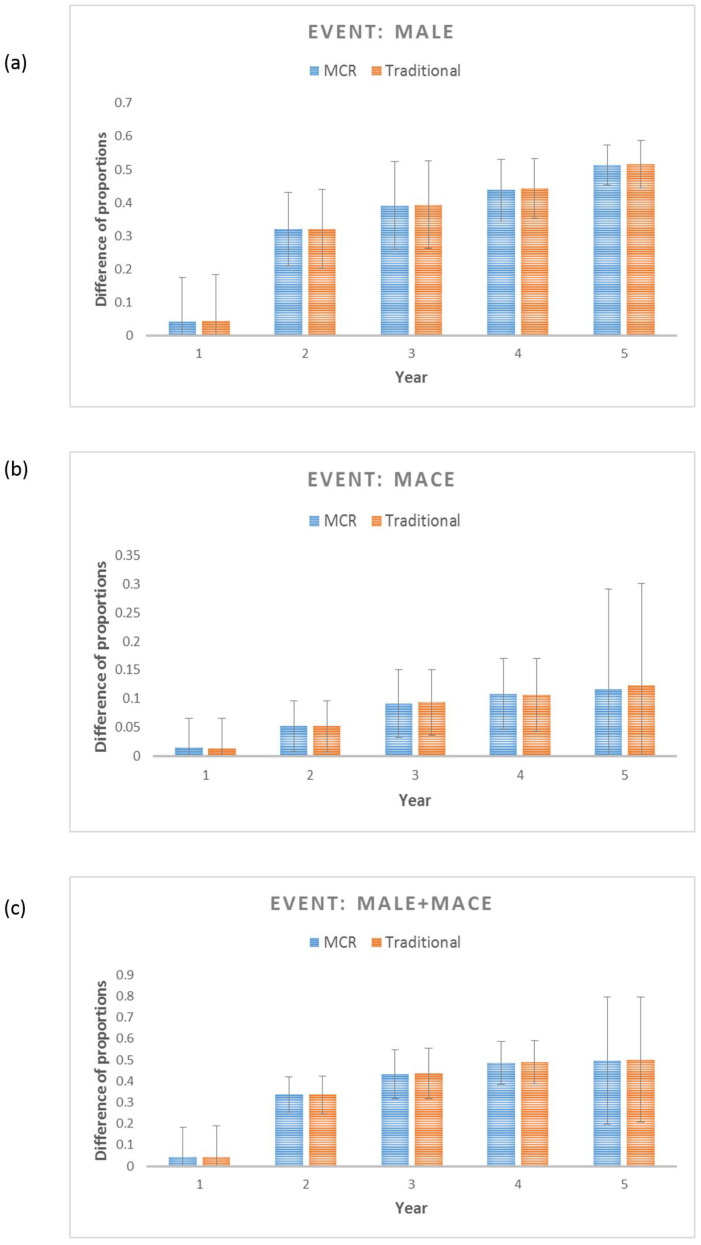
Calibration plots for (**a**) MALEs, (**b**) MACEs, and (**c**) MALEs + MACEs showing the difference between the observed and predicted survival probability for the proposed MCR-based prognostic model (with an MCR score) and the traditional model (only traditional variables without an MCR score). Calibration for each of the models was conducted using 10-fold cross-validation (CV), and each bar shows an average of the probability difference over 10 models for each CV.

**Figure 5 biomedicines-12-01374-f005:**
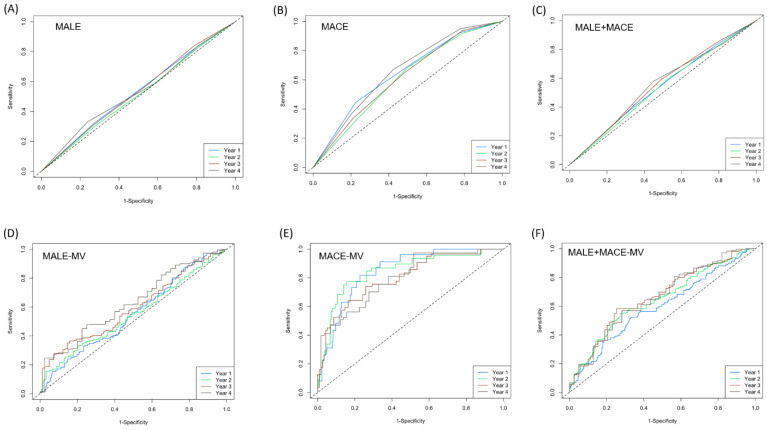
ROC plots for 12 months, 24 months, 36 months, and 48 months using univariate models and multivariate adjusted models with CHA_2_DS_2_-VASc as the predictor. (**A**–**C**) Univariate models for MALEs, MACEs, and MALEs + MACEs, respectively. (**D**–**F**) Multivariate adjusted models for MALEs, MACEs, and MALEs + MACEs, respectively.

**Table 1 biomedicines-12-01374-t001:** Characteristics of patients with peripheral artery disease.

Characteristic (Units)	Measurement N = 503
Age (years)	70.77 ± 12.39
Sex male	326 (64.81)
female	177 (35.19)
BMI (kg/m^2^)	23.97 ± 3.91
CHF (C)	238 (47.32)
HTN	403 (86.68)
DM	376 (74.75)
Stroke/TIA	91 (18.09)
Vascular disease	503 (100)
HPL	241 (47.91)
SMK	195 (38.77)
CAD	263 (52.29)
CABG	53 (10.54)
PCI	239 (47.51)
Old MI	79 (15.71)
COPD	21 (4.17)
CKD	319 (63.42)
HD/PD	181 (35.98)
Cr (mg/dL)	3.26 ± 3.04
Af	120 (23.86)
Imd	21 (4.03)
HbA1C (%)	7.32 ± 1.86
Cholesterol (mg/dL)	149.65 ± 39.42
LDL (mg/dL)	83.37 ± 33.35
HDL (mg/dL)	42.94 ± 15.21
TG (mg/dL)	130.87 ± 83.79
Glu (mg/dL)	145.60 ± 69.49
TG/HDL	3.69 ± 3.86
ASA	385 (76.54)
Clopidgrel	427 (84.89)
Cilostazol	301 (59.84)
Pentoxyphilline	19 (0.19)
Direct oral anticoagulant (DOAC)	73 (14.51)
ACEIARB	220 (43.74)
Statin	283 (56.26)
Betablocker	189 (37.57)
CCB	201 (39.96)
Insulin	106 (21.07)
Rutherford = 1	0 (0)
Rutherford = 2	0 (0)
Rutherford = 3	0 (0)
Rutherford = 4	130 (25.84)
Rutherford = 5	316 (62.82)
Rutherford = 6	57 (11.33)
Target vessel CIA	41 (8.15)
Target vessel EIA	45 (8.95)
Target vessel CFA	27 (5.37)
Target vessel SFA	285 (56.66)
Target vessel ATA	248 (49.30)
Target vessel popliteal	107 (21.27)
Target vessel peroneal artery	96 (19.09)
Target vessel tibiofibular TP trunk	64 (12.72)
Target vessel PTA	196 (38.97)
Target vessel DPA	15 (2.98)
Target vessel plantar artery	23 (4.57)

All measures are depicted as the mean ± standard deviation or n (%). BMI: body mass index; CHF: congestive heart failure; HTN: hypertension; DM: diabetes mellitus; TIA: transient ischemic attack; HPL: hyperlipidemia; SMK: smoking status; CAD: coronary artery disease; CABG: coronary artery bypass graft; PCI: percutaneous coronary intervention; MI: myocardial infarction; COPD: chronic obstructive pulmonary disease; CKD: chronic kidney disease; HD/PD: hemodialysis/peritoneal dialysis; Cr: creatinine; Af: atrial fibrillation; Imd: immune-related disease; HbA1C: hemoglobin A1C; LDL: low-density lipoprotein; HDL: high-density lipoprotein; TG: triglyceride; Glu: glucose; ASA: acetylsalicylic acid; ACEIARB: angiotensin-converting enzyme inhibitor (ACEI)/angiotensin receptor blocker (ARB); CCB: calcium channel blocker; Rutherford: Rutherford classification; CIA: common iliac artery; EIA: external iliac artery; CFA: common femoral artery; SFA: superficial femoral artery; ATA: anterior tibial artery; tibiofibular TP trunk: tibiofibular tibioperoneal trunk; PTA: posterior tibial artery; DPA: dorsalis pedis artery.

**Table 2 biomedicines-12-01374-t002:** Characteristics of 503 PAD patients (N = 503) divided into risk groups based on their MCR score.

Variable	Score = 3 (N = 100)	Score = 4 (N = 115)	Score = 5 (N = 140)	Score = 6 (N = 148)	*p* Value
Age	59.14 ± 12.07	67.76 ± 11.06	72.2 ± 10.54	79.60 ± 6.800	<0.0001 *
Sex (Male)	86 (86)	93 (80.87)	84 (60)	63 (42.57)	<0.0001 *
BMI	23.96 ± 3.85	24.48 ± 4.414	23.59 ± 3.503	23.93 ± 3.881	0.341
CHF (C)	13 (13)	50 (43.48)	67 (47.86)	108 (72.97)	<0.0001 *
HTN	52 (52)	101 (87.82)	136 (97.14)	147 (99.32)	<0.0001 *
DM	44 (44)	89 (77.39)	107 (76.43)	136 (91.89)	<0.0001 *
Stroke (S)/TIA	3 (3)	7 (6.09)	28 (20)	53 (35.81)	<0.0001 *
Vascular disease	100 (100)	115 (100)	140 (100)	148 (100)	1
Hyperlipidemia	36 (36)	51 (44.53)	69 (49.28)	85 (57.43)	0.008 *
SMK (smoking)	63 (63)	60 (52.17)	44 (31.43)	28 (18.92)	<0.0001 *
CR disease	28 (28)	60 (52.17)	85 (60.71)	90 (60.81)	<0.0001 *
Coronary artery bypass graft (CABG)	3 (3)	15 (13.04)	15 (10.71)	20 (13.51)	0.022 *
Percutaneous coronary intervention (PCI)	22 (22)	57 (49.57)	77 (55)	83 (56.08)	<0.0001 *
Old myocardial infarction (MI)	5 (5)	19 (16.52)	26 (18.57)	29 (19.59)	0.004 *
COPD	2 (2)	2 (1.74)	9 (6.43)	8 (5.4)	0.168
CKD	35 (35)	70 (60.87)	109 (77.86)	105 (70.95)	<0.0001 *
HD/PD	25 (25)	42 (36.52)	55 (39.29)	59 (39.86)	0.069
Cardiac rehabilitation (Cr) score	2.65 ± 3.24	3.493 ± 3.569	3.536 ± 2.898	3.226 ± 2.553	0.136
Atrial fibrillation (Af)	10 (10)	19 (16.52)	40 (28.47)	51 (34.46)	<0.0001 *
Immune-related disease (Imd)	10 (10)	3 (2.61)	4 (2.86)	4 (2.70)	0.033 *
HbA1C	7.013 ± 1.928	7.675 ± 2.129	7.285 ± 1.600	7.298 ± 1.799	0.072
Cholesterol	163.65 ± 40.72	149.50 ± 42.07	147.81 ± 40.29	142.05 ± 32.99	0.0003 *
LDL	93 ± 34.675	82.03 ± 32.08	83.22 ± 36.68	78.01 ± 28.69	0.006 *
HDL	43.76 ± 18.288	43.02 ± 16.36	41.53 ± 12.31	43.770 ± 14.486	0.542
TG	147.31 ± 99.34	131.71 ± 93.15	129.2 ± 72.41	120.67 ± 73.13	0.106
Glu	135.29 ± 65.37	154.71 ± 80.44	147.75 ± 68.83	143.44 ± 63.002	0.216
Medications					
ASA	79 (79)	88 (76.52)	111 (79.29)	107 (72.30)	0.504
Clopidgrel	76 (79)	97 (84.35)	124 (88.57)	130 (87.84)	0.0428
Cilostazol	62 (62)	77 (66.96)	79 (56.43)	83 (56.08)	0.242
Pentoxyphilline	0 (0)	0 (0)	0 (0)	1 (0.67)	1
Direct oral anticoagulant (DOAC)	15 (15)	11 (9.56)	24 (17.14)	23 (15.54)	0.349
ACEIARB	35 (35)	54 (46.96)	67 (47.86)	64 (43.24)	0.204
Statin	57 (57)	63 (54.78)	81 (57.86)	82 (55.41)	0.958
Betablocker	22 (22)	48 (41.74)	58 (41.43)	61 (41.22)	0.003 *
CCB	37 (37)	43 (37.39)	66 (47.14)	55 (37.16)	0.251
Insulin	13 (13)	26 (22.61)	28 (20)	39 (26.35)	0.076
Rutherford classification					
1	0 (0)	0 (0)	0 (0)	0 (0)	1
2	0 (0)	0 (0)	0 (0)	0 (0)	1
3	0 (0)	0 (0)	0 (0)	0 (0)	1
4	36 (36)	28 (24.34)	34 (24.29)	32 (21.62)	0.079
5	55 (55)	72 (62.61)	86 (61.43)	103 (69.59)	0.129
6	9 (9)	15 (13.04)	20 (14.29)	13 (8.78)	0.394
Target vessel					
CIA	8 (8)	11 (9.56)	15 (10.71)	7 (4.73)	0.249
EIA	11 (11)	11 (9.56)	13 (9.29)	10 (6.76)	0.677
CFA	10 (10)	4 (3.48)	8 (5.71)	5 (3.38)	0.128
SFA	42 (42)	59 (51.30)	86 (61.43)	98 (66.22)	0.0007 *
ATA	49 (49)	62 (53.91)	70 (50)	67 (45.27)	0.581
Popliteal	19 (19)	19 (16.52)	33 (23.57)	36 (24.32)	0.375
Peroneal artery	12 (12)	22 (19.13)	27 (19.29)	35 (23.65)	0.147
Tibiofibular TP trunk	9 (9)	6 (5.21)	23 (16.43)	26 (17.57)	0.005 *
PTA	44 (44)	46 (40)	50 (35.71)	56 (37.84)	0.613
DPA	4 (4)	4 (3.48)	3 (2.14)	4 (2.70)	0.825
Plantar artery	2 (2)	8 (6.96)	8 (5.71)	5 (3.38)	0.269

Number of abnormal parameters (≤3, 4, 5, and ≥6). BMI: body mass index; CHF: chronic heart failure; HTN: hypertension; DM: diabetes mellitus; TIA: transient ischemic attack; HPL: hyperlipidemia; SMK: smoking status; CAD: coronary artery disease; CABG: coronary artery bypass graft; PCI: percutaneous coronary intervention; MI: myocardial infarction; COPD: chronic obstructive pulmonary disease; CKD: chronic kidney disease; HD/PD: hemodialysis/peritoneal dialysis; Cr: creatinine; Af: atrial fibrillation; Imd: immune-related disease; HbA1C: hemoglobin A1C; LDL: low-density lipoprotein; HDL: high-density lipoprotein; TG: triglyceride; Glu: glucose. *: significant with *p*-value < 0.05.

**Table 3 biomedicines-12-01374-t003:** Performance of the MCR score as a predictor of MACEs using 503 patients with peripheral artery disease.

Events	Low Risk (N = 100)	Moderate Risk (N = 115)	High Risk (N = 140)	Very High Risk (N = 148)
Major adverse cardiovascular events (MACEs)				
#MACEs (%)	3 (3)	13 (11.30)	16 (11.43)	17 (11.49)
Crude HR (95% CI)	1	3.52 (1.00–12.12)	4.18 (1.22–14.36)	5.08 (1.49–17.36)
*p* value		0.048 *	0.023 *	0.009 *
Multivariate adjusted HR (95% CI)	1	2.57 (0.55–9.31)	3.51 (0.78–15.86)	4.67 (1.03–21.09)
*p* value		0.231	0.102	0.045 *
Major adverse limb events (MALEs)				
# MALEs (%)	31 (31)	49 (42.60)	63 (45)	50 (33.78)
Crude HR (95% CI)	1	1.33 (0.85–2.09)	1.55 (1.01–2.38)	1.21 (0.77–1.89)
*p* value		0.213	0.046 *	0.398
Multivariate adjusted HR (95% CI)	1	1.29 (0.81–2.33)	1.66 (1.04–3.2)	1.34 (0.79–2.94)
*p* value		0.30	0.039 *	0.249
Major adverse limb and cardiac events (MALEs + MACEs)				
#MALEs + MACEs (%)	34 (34)	57 (49.57)	70 (50)	63 (42.57)
Crude HR (95% CI)	1	1.37 (0.89–2.09)	1.58 (1.05–2.37)	1.41 (0.93–2.14)
*p* value		0.145	0.029 *	0.107
Multivariate adjusted HR (95%CI)	1	1.35 (0.85–2.13)	1.6 (1.01–2.53)	1.46 (0.92–2.30)
*p* value		0.204	0.045 *	0.108

Multivariate model adjusted by factors that were found significant in Appendix A for MACEs, MALEs, and MALEs + MACEs. Multivariate models were adjusted by coronary artery disease, coronary artery bypass graft (CABG), chronic obstructive pulmonary disorder, myocardial infarction, chronic kidney disease, hemodialysis, peritoneal dialysis, creatinine, and atrial fibrillation. *: significant with *p*-value < 0.05.

**Table 4 biomedicines-12-01374-t004:** Averages and standard deviations of c-indices from 10-fold cross-validation.

	MALEs		MACEs		MALEs + MACEs	
	Avg. C-Index	Std. Dev. C-Index	Avg. C-Index	Std. Dev. C-Index	Avg. C-Index	Std. Dev. C-Index
Crude MCR model	0.54	0.009	0.63	0.02	0.54	0.009
Multivariate-adjusted MCR model	0.57	0.009	0.81	0.014	0.56	0.009
Traditional model	0.55	0.01	0.81	0.01	0.54	0.007

MCR: modified cumulative risk; MALEs: major adverse limb events; MACEs: major adverse cardiovascular events. Multivariate model adjusted by hyperlipidemia, smoking status, chronic obstructive pulmonary disease, chronic kidney disease, hemodialysis/peritoneal dialysis, creatinine, atrial fibrillation, immune-related disease, hemoglobin, cholesterol, low-density lipoprotein, high-density lipoprotein, triglyceride, and glucose levels.

## Data Availability

The original contributions presented in this study are included in this article/Appendix A. Further inquiries can be directed to the corresponding author.

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
