# Peer review of "Assessing the Suitability of CHA2DS2-VASc for Predicting Adverse Limb Events and Cardiovascular Outcomes in Peripheral Artery Disease Patients with Percutaneous Transluminal Angioplasty: A Retrospective Cohort Study"

_biomedicines, 2024, doi:10.3390/biomedicines12061374_

Round 1

Reviewer 1 Report

Comments and Suggestions for Authors

The CHA2DS2-VASc score is a clinical prediction tool initially developed for assessing stroke risk in patients with atrial fibrillation (AF). However, researchers have also explored its potential application in predicting adverse limb events and cardiovascular outcomes in patients with peripheral artery disease. Studies have indicated that higher CHA2DS2-VASc scores are associated with an increased risk of adverse limb events and cardiovascular outcomes in PAD patients. However, further research is needed to validate its utility specifically in this patient population and to determine if modifications or additional factors are necessary for optimal risk prediction.

This study presents the possible effectiveness of the CHA2DS2-VASc score in predicting major adverse limb events (MALE) and major adverse cardiovascular events (MACE) in patients with peripheral artery disease (PAD). The abstract highlights the gap in the literature regarding the evaluation of the CHA2DS2-VASc score for predicting adverse outcomes in PAD patients and aims to address a critical need for risk stratification tools in this patient population.

Previously, others factors have been associated to an increased risk of MACE in patients with CHA2DS2-VASc score [1-3] as well as peripheral arteriopathy [3]. I recommend reviewing these articles and considering their inclusion as references.

[1] Miao B, Hernandez AV, Roman YM, Alberts MJ, Coleman CI, Baker WL. Four-year incidence of major adverse cardiovascular events in patients with atherosclerosis and atrial fibrillation. Clin Cardiol. 2020;43(5):524-531. doi:10.1002/clc.23344

[2] Harb SC, Wang TKM, Nemer D, Wu Y, Cho L, Menon V, Wazni O, Cremer PC, Jaber W. CHA2DS2-VASc score stratifies mortality risk in patients with and without atrial fibrillation. Open Heart. 2021, 8(2):e001794. doi: 10.1136/openhrt-2021-001794

[3] Moltó-Balado P, Reverté-Villarroya S, Monclús-Arasa C, Balado-Albiol MT, Baset-Martínez S, Carot-Domenech J, Clua-Espuny JL. Heart Failure and Major Adverse Cardiovascular Events in Atrial Fibrillation Patients: A Retrospective Primary Care Cohort Study. Biomedicines. 2023 Jun 26;11(7):1825. doi: 10.3390/biomedicines11071825.

In spite of the study stratified PAD patients into four risk groups based on their CHA2DS2-VASc scores and the results suggest that the CHA2DS2-VASc score significantly may predict MACE particularly in moderate- to very-high-risk groups,  there is not enough content regarding its utility in daily clinical practice (e.g., changes in secondary prevention guidelines, indications for surgery, etc.) potentially guiding personalized treatment strategies, since it deals with patients already diagnosed with PAD.

Some specific suggestions related to formatting are:

1/ Check the Abstract and rewrite possible mistakes: As I believe there are errors in the description of CHA2DS2-VASc as a significant predictor.

2/ Consider removing “legitimate”: with the current results, the adjective "legitimate" in point 3.4 should be considered for removal.

3/ Adjust for presence of AF: the presence or absence of atrial fibrillation (AF) is a fundamental factor for which the results should be adjusted or stratified, as its prevalence increases with the CHA2DS2-VASc score. Its treatment, along with the Time in Therapeutic Rank (TTR%) in patients treated with vitamin K antagonists, is crucial to consider when adjusting the results based on whether AF is a comorbidity present in patients with PAD.

4/ The description of the results is too long. It would benefit from rearranging and pointing out the results with greater impact on clinical practice.

Comments on the Quality of English Language

Minor

Author Response

Dear Reviewer

We would like to resubmit our revised manuscript titled “Assessing the suitability of CHA2DS2-VASc for predicting adverse limb events and cardiovascular outcomes in Peripheral Artery Disease patients with Percutaneous Transluminal Angioplasty: A retrospective cohort study”, by Cheng et al., which we hereby submit to be considered for publication in Biomedicines (biomedicines-3008958).

We appreciated the suggestions from the reviewers and the editor. The comments were favorable and encouraging, and a point-to-point response is provided .

This manuscript, as submitted or its essence in another version, is not under consideration for publication elsewhere, and has no overlap with another existing publication or submission. The authors declare no financial or conflict of interest. All authors have made substantive contributions to the study and endorsed the data and conclusions. We believe that this work may be of great interest to the readers of Biomedicines.

Thanking You

Sincerely

Chien-Shan Chiu, M.D., Ph.D.

Department of Dermatology, Taichung Veterans General Hospital, 1650 Section 4 Taiwan   Boulevard, Xitun District, Taichung 40705, Taiwan

Reviewer 2 Report

Comments and Suggestions for Authors

In this interesting study, Cheng al. evaluated the role of CHA2DS2-VASc as a potential risk score for MACE and MALE in patients with lower extremity PAD who underwent PTA. The study is well conducted and presented but some issues should be addressed to improve the study. Below are some comments and suggestions:

1.     the study design should be explained in the title, abstract and materials and methods.

2.     What were the indications for PTA treatment in the enrolled patients? Please explain it in the methods, abstract and title.

3. In an interesting study, the authors demonstrated how critical vascular events affecting the lower limbs in subjects with PAD and stenosis greater than 70% were largely attributable not to progression of the stenosis but to thrombotic phenomena [Narula N, Dannenberg AJ, Olin JW, Bhatt DL, Johnson KW, Nadkarni G, Min J, Torii S, Poojary P, Anand SS, Bax JJ, Yusuf S, Virmani R, Narula J. Pathology of Peripheral Artery Disease in Patients With Critical Limb Ischemia. J Am Coll Cardiol. 2018 Oct 30;72(18):2152-2163. doi: 10.1016/j.jacc.2018.08.002. Epub 2018 Aug 27. PMID: 30166084.]. In this sense, a recent review on the role of coagulation in patients with PAD has highlighted the importance of thrombotic phenomena in patients with arterial disease of the lower limbs [Miceli G, Basso MG, Rizzo G, Pintus C, Tuttolomondo A. The Role of the Coagulation System in Peripheral Arterial Disease: Interactions with the Arterial Wall and Its Vascular 
Microenvironment and Implications for Rational Therapies. Int J Mol Sci. 2022 Nov 29;23(23):14914. doi: 10.3390/ijms232314914. PMID: 36499242; PMCID: PMC9739112.
]. Considering that your study aims to test the use of an already validated score for thromboembolic phenomena, I suggest carrying out a separate subanalysis for patients with stenosis greater than 70% and evaluating whether the CHADVASC2 is predictive of MALE in this subgroup of patients. I recommend commenting on the two bibliographical references in the discussion.

Comments on the Quality of English Language

English should be improved to be more fluent. 

Author Response

Dear Reviewer

We would like to resubmit our revised manuscript titled “Assessing the suitability of CHA2DS2-VASc for predicting adverse limb events and cardiovascular outcomes in Peripheral Artery Disease patients with Percutaneous Transluminal Angioplasty: A retrospective cohort study”, by Cheng et al., which we hereby submit to be considered for publication in Biomedicines (biomedicines-3008958).

We appreciated the suggestions from the reviewers and the editor. The comments were favorable and encouraging, and a point-to-point response is provided as appended below.

This manuscript, as submitted or its essence in another version, is not under consideration for publication elsewhere, and has no overlap with another existing publication or submission. The authors declare no financial or conflict of interest. All authors have made substantive contributions to the study and endorsed the data and conclusions. We believe that this work may be of great interest to the readers of Biomedicines.

Thanking You

Sincerely

Chien-Shan Chiu, M.D., Ph.D.

Department of Dermatology, Taichung Veterans General Hospital, 1650 Section 4 Taiwan   Boulevard, Xitun District, Taichung 40705, Taiwan

Round 2

Reviewer 1 Report

Comments and Suggestions for Authors

The manuscript has been checked according the requirements and may be accepted in present form

Reviewer 2 Report

Comments and Suggestions for Authors

I do not have any other comments. Thank you for the answers.